# Insights into Predictors of Vaccine Hesitancy and Promoting Factors in Childhood Immunization Programs—A Cross-Sectional Survey in Cameroon

**DOI:** 10.3390/ijerph19052721

**Published:** 2022-02-26

**Authors:** Jonas Kemeugni Ngandjon, Thomas Ostermann, Virgile Kenmoe, Alfred Laengler

**Affiliations:** 1Institute of Pediatric, Faculty of Health, University of Witten/Herdecke, 58448 Witten, Germany; 2Department for Psychology and Statistics, Faculty of Health, University of Witten/Herdecke, 58448 Witten, Germany; thomas.ostermann@uni-wh.de; 3Department for Human Medicine , Faculty of Health, Université des Montagnes, Bangangté BP 208, Cameroon; jboom@gmx.de; 4Department for Human Medicine, Institute of Pediatric, Faculty of Health, University of Witten/Herdecke, 58448 Witten, Germany; alfred.laengler@uni-wh.de

**Keywords:** vaccine hesitancy, vaccine acceptance, childhood vaccination, knowledge of vaccine-preventable diseases, vaccination status, COVID-19, health education

## Abstract

Background: Vaccination is claimed to be a key intervention against the COVID-19 pandemic. A major challenge today is to increase vaccine acceptance as vaccine hesitancy has delayed the eradication of polio. This study aimed to identify predictors associated with vaccine acceptance in the context of the Expanded Program on Immunization among parents of children between the ages of 12 to 23 months in the Foumbot district, Cameroon. Methods: The design of this study is a cross-sectional survey. A total of 160 mothers of children between the ages of 12 to 23 months were selected using a simple random sampling technique. A pre-tested structured questionnaire was used for data collection. Incomplete vaccination status was considered “vaccine hesitancy”. Data was analyzed along with 95% confidence intervals and the *p*-value < 0.05. The results showed 60% vaccine acceptance and 40% vaccine hesitancy. Factors such as age-appropriate vaccination, knowledge of vaccine-preventable diseases (VPD), and religion were associated with vaccine acceptance. Conclusion: Poor knowledge of VPDs is a matter of concern as it contributes to vaccine hesitancy. The study findings provide the basis to heighten health education, the public perceived threat of the VPDs, and the consequences if no measures are taken to ensure health.

## 1. Background

Early vaccination success and necessities to eradicate communicable diseases brought widespread vaccine acceptance, and immunization programs have significantly decreased the incidence of numerous diseases in many parts of the world [1,2]. As the COVID-19 pandemic is threatening the world, the advent of COVID-19 vaccines was widely considered a key intervention in disease control. In addition to the Expanded Program on Immunization (EPI), which is a free immunization program to reach each child with polio, measles, tetanus, diphtheria, pertussis, and tuberculosis vaccines, governments in low-income countries have also launched the COVID-19 immunization program [2,3,4]. Reports from the World Health Organization (WHO) show that delays in polio eradication are linked to vaccine hesitancy among parents [2]. Therefore, the success of immunization programs depends not only on vaccines supply but also on vaccine acceptance. In 2020, many countries had still not achieved 80% polio herd immunity (three polio vaccine doses) despite the availability of vaccines: Cameroon (70%), the Democratic Republic of Congo (59%), Nigeria (57%), and Central African Republic (46%) [1]. During the COVID-19 pandemic, as COVID-19 vaccines arrived in Africa, Nigeria administered 48%, Cameroon 23%, and the Democratic Republic of Congo administered fewer than 5% of the vaccine doses that they received [5]. By December 2021, COVID-19 vaccine hesitancy was 84% among Cameroonians, and only 2.6% of the population had received two COVID-19 vaccine doses [1,6]. The sole provision of vaccines does not guarantee vaccine acceptance. Due to diverse reasons and factors, vaccine hesitancy remains a concern on the African continent, although immunization programs (EPI and COVID-19 vaccines) are free of charge. Vaccine hesitancy refers to a delay in acceptance or refusal of an available vaccine. Many vaccine-hesitant parents believe that vaccines contain suspicious products, which could endanger child health [4]. The feeling that sows doubt and mistrust, as well as the spread of misinformation, both lead to a community of vaccine hesitancy. Studies of recent outbreaks of vaccine-preventable diseases (VPDs) have reported that vaccine hesitancy among parents is linked to the outbreak of VPDs throughout African countries and internationally [7,8,9]. In 2013, outbreaks of polio were reported in Central Africa, particularly in Cameroon, where it was reported in nine districts, including the Foumbot district [10]. Despite the EPI in this district, children with polio had not received any vaccine by the age of two. A major challenge today is to increase vaccine acceptance.

Although evidence on encouraging vaccination, in general, is useful in the context of outbreaks, vaccine acceptance poses an enormous challenge. Understanding the process of vaccine acceptance can serve as an early warning system to take action to avoid declining vaccine confidence and acceptance. This study offers a baseline of confidence levels to assess vaccine acceptance among parents and help identify where further confidence-building is required to increase adoption of childhood vaccination and new vaccines such as COVID-19 vaccines.

However, data on vaccine acceptance from African countries, mainly from Central Africa, is substantially limited. This indicates a need for specific research in these regions to explore the factors that contribute to parental vaccine hesitancy. Evidence from western countries will not be applicable here due to the significant differences in social and cultural contexts. Therefore, this study aimed to identify, during the EPI, the factors influencing complete childhood vaccination by sociodemographics and knowledge on vaccination among parents in the Foumbot district in Cameroon to understand the gap in formulating a comprehensive immunization program.

## 2. Methods

### 2.1. Study Area

The Foumbot district is a rural area of about 1000 km^2^. It is located in the Noun Division, West Region of Cameroon. The population estimate is 62,776 inhabitants (from the 2013 Census), the majority being Bamum. The predominant religion is Islam. Farming is the main occupation. The district hosts the most important fresh food market in Cameroon. The EPI is provided free of charge at health facilities. In 2013, a polio outbreak was reported in the district [10].

### 2.2. Study Design

A cross-sectional survey of parents of children aged 12 to 23 months was conducted from 1st July to 31st October 2014. The EPI coverage and the factors associated with childhood vaccination were assessed. Only the parents of the children were interviewed. Survey participants were selected randomly according to coverage cluster survey sampling [11,12,13,14]. Vaccination coverage was evaluated by means of the vaccination booklet and the EPI registers. 

### 2.3. Sample Size Determination

Using the sample size calculation methodology presented in the WHO Immunization Coverage Cluster Survey Reference Manual (WHO/IVB/04.23), the sample size required was determined using the coverage of 64% obtained in the western region, a precision of ± 10%, a type 1 error of 10%, and a design effect of 1.5, in conformity with the standard WHO methodology [12,13]. Thus, the calculated minimum number of children required was 147.

### 2.4. Participants

During the investigation, data from 160 children and 160 parents were randomly collected in the district. The sampling process was performed according to the simple random sample method [11,12]. The first household was randomly chosen from each selected cluster. Each household was chosen randomly, such that each household had the same probability in the cluster of being chosen during the sampling process [15]. 

### 2.5. Data Collection

The data was collected using a structured questionnaire. The questionnaire covers sociodemographic factors and parents’ knowledge of VPDs. Prior to data collection, the questionnaire was pretested on a 5% sample of a similar population. The vaccination status of the child was established by the records in the vaccination booklet, the records in the EPI registers, or by the presence of scar in the case of the BCG vaccine. 

### 2.6. Independent Variables

The independent variables included sociodemographic characteristics and the parent’s knowledge regarding vaccination and VPDs. The sociodemographic variables included age group (<25, 25–34, >34), education (no formal education, primary, secondary, university), occupation (formal sector, informal sector), marital status (married, single, divorced, and widowed), religion (Catholic, Protestant, Muslim, and Animist). To collect data on knowledge about vaccination and VPDs, parents were asked if they knew the reason for vaccinating (prevent the occurrence of diseases, decrease the severity of diseases, no idea), had knowledge of VPDs (good knowledge, poor knowledge), knew age-appropriate vaccination (at birth, no idea, other date), were aware of the vaccination schedule (be aware), and if they retained the vaccination booklet (keep). The response was recorded as ‘Yes/No’. 

### 2.7. Outcome Variable

A close-ended question inquiring whether the parent has completed the EPI was used as the outcome variable. The outcome variable for this study was complete vaccination. Complete vaccination was defined as a child who received one dose of BCG and measles and three doses of OPV before the age of two. Children missing one dose of these vaccines were defined as incomplete vaccination. Complete vaccination was considered ‘vaccine acceptance’, whereas incomplete vaccination and no vaccination were considered ‘vaccine hesitancy’. 

### 2.8. Data Processing and Analysis

The data from the interview was coded and entered into a computer database using *Microsoft Office Excel 2010*. Descriptive statistics were performed by means of the *statistical software program SPSS*. The Fisher test was used, and the odds ratios (OR) were calculated along with 95% confidence intervals (CI). A bivariate analysis was conducted to examine the association of vaccination status with each independent variable. Based on this, only independent variables that had significant bivariate associations with vaccination status were considered for the multivariate logistic regression. The stepwise multivariate analysis was used. Models were generated according to the type “forward selection”. Akaike’s information criterion (AIC) and Schwarz’s Bayesian information criteria (BIC) were used to inform the selection of models and to assess the goodness of fit. The AIC and BIC values were compared in successive models, and the model with the lowest value was considered the best-fit model [16]. In addition, the likelihood ratio test and the area under receiver operating characteristics (AUC) curve were used to assess the predictive quality of the model.

### 2.9. Ethical Clearance

This study obtained authorization to be carried out from the Faculty of Medicine of the Université des Montagnes and the health authorities of the Foumbot district. Verbal informed consent was required for each participant prior to the administration of the questionnaire.

## 3. Results

### 3.1. Prevalence of Vaccine Acceptance

Table 1 present the vaccination coverage in the study area. It appears that 96 (60%) of the 160 respondents completed the EPI. About 62 (38.75%) of the respondents had not completed the EPI, and 2 (1.25%) of the respondents had not started with the EPI. Therefore, 64 (40%) of respondents indicated vaccine hesitancy. The dropout rate between the initial vaccine of BCG and the final vaccine of measles was 36%.

### 3.2. Barriers in Vaccine Acceptance

Table 2 present the reasons for missing vaccination among parents whose children had incomplete vaccination status. Eleven (17.2%) of the sixty-four vaccine hesitant parents declared a lack of information regarding childhood vaccination. Additionally, 11 (17.2%) reported a lack of confidence in the EPI, whereas 42 (65.6%) reported poor quality of vaccination services.

### 3.3. Sociodemographic in Vaccine Acceptance

Table 3 show the results of the bivariate analysis. The factors marital status, religion, knowledge of VPDs, age-appropriate vaccination, and being aware of the vaccination schedule were significantly associated with complete vaccination at *p* < 0.05. The odds of receiving a complete vaccination among children whose parents were single were 3.23 (OR = 3.23; 95% CI: 1.03, 10.09) times higher than children whose parents were married. Muslim parents had 0.27 (OR = 0.27; 95% CI: 0.12, 0.61) times lower odds of completely vaccinating their children than parents from other religions. Children whose parents started vaccination at birth were 2.5 (OR = 2.5; 95% CI: 1.12, 5.55) times more likely of being completely vaccinated compared to children whose parents started later. Parents who were aware of the vaccination schedule were 3.52 (OR = 3.52; 95% CI: 1.43, 8.65) times more likely to have completely vaccinated their child compared to parents who were not aware of it. The odds of being completely vaccinated among children whose parents had good knowledge of VDPs were 3.08 (OR = 3.08; 95% CI: 1.21, 7.85) times higher than children whose parents had poor knowledge of VPDs.

### 3.4. Multivariate Logistic Regressions

Table 4 present the results of the stepwise method based on forward selection to purposely assess the contribution of each independent variable towards the prediction of complete childhood vaccination. The effect of the variables (religion, age-appropriate vaccination, knowledge of VPDs) was statistically significant (that is with Sig. < 0.05). This means that the variables found in Table 4 contribute significantly to the explanatory power of parents’ decision to complete vaccination. However, the following variables (marital status and being aware of the vaccination schedule) were dropped from the final model with the stepwise method of forward selection since they did not contribute towards the prediction of complete vaccination in Appendix A.

Table 5 depict the model fitting information for the inclusion of the three variables that were found significant in Table 4. The final model with the least log-likelihood (56.16), AIC (64.16), BIC (76.46), and a degree of freedom of 3 was highly significant at the 5% level of significance. This means the final model based on the likelihood ratio test confirms that the multivariate logistic regression comprising of the three predictors fits the data better and is more effective than by chance towards the prediction of childhood vaccination status. The predictive utility of the final model was fair, with an AUC of 0.73.

## 4. Discussion

This study examined the association and influence of sociodemographic factors and knowledge of VPDs on complete childhood vaccination. The complete childhood vaccine acceptance was 60% [Table 1]. This was high as compared to the three Cameroonian regions West (54%), Adamawa (38%), and North (36%) [17]. In total, 65.6% of the vaccine-hesitant parents mentioned barriers linked to the quality of vaccination services [Table 2]. Similar results were reported in studies carried out in Bangladesh, India, the Philippines, Ethiopia, and Malawi [18,19]. In these countries, shortcomings were identified at the interface between the vaccination provider and the parents. Thus, the pools of unvaccinated children increased. In 2015, Cameroon was still classified as having an acute shortage of health personnel [20]. Researchers in Vietnam observed that the advice of a healthcare professional was the major factor that changed the view of parents who had previously refused vaccination or had delayed vaccination [21,22]. Therefore, contact with a healthcare professional is an important factor in vaccine acceptance. The results of this study showed that age-appropriate vaccination was significantly associated with complete childhood vaccination. Once starting the vaccination program at birth, the likelihood to complete the EPI increased by 2.5 [OR = 2.5] [Table 3]. A study performed in 2000 in a rural community of the Edo State in Nigeria showed that early vaccination of children increases the awareness of the parents towards VPDs and vaccination [23]. Administering the first vaccine at birth is an important step for building confidence in medical interventions and raising parental awareness about VPDs and the role that the vaccination can play in promoting child health. However, we found that 36% of the children had not completed the vaccination program, although 98% had received the BCG vaccine at birth [Table 1]. This study revealed that 17.2% of the vaccine-hesitant parents had concerns about the quality of information on the vaccination. Additionally, 17.2% did not have trust in the vaccine [Table 2]. This indicates that the awareness towards VPDs was not reached, and the confidence in medical interventions was not established. This study also revealed that religion was significantly associated with complete childhood vaccination [Table 3]. Data from India, Kenya, and Nigeria showed that vaccination in countries with traditional societies was more often associated with religious leaders as they are key stakeholders in disease prevention [4,24,25,26,27,28,29,30]. In these countries, vaccine hesitancy was driven by religious leaders as they called for a boycott of polio vaccination campaigns citing safety concerns with vaccines so that medical interventions intersected with cultural perception [31,32,33,34,35]. Due to past experiences relating to the slave trade and the colonization of Africa, vaccination has become suspicious as it is often linked to the conspiracy theory that western countries want to depopulate the African continent. Furthermore, many people believe in vaccine-induced female infertility [4,24,26,27,34,35,36,37]. These previous studies showed that vaccine acceptance was not generated by religious principles but by how health information was conveyed, perceived, and understood. The results of this study showed that knowledge of VPDs was a significant predictor to complete the EPI. Parents with good knowledge of VPDs [OR = 3.08] had a higher likelihood of completing the EPI. This finding is similar to an international study carried out in five African countries (Democratic Republic of Congo, Benin, Uganda, Malawi, and Mali), which reported that during the COVID-19 pandemic, individuals who perceived taking the COVID-19 vaccine as important to protect themselves had the highest vaccine acceptance odd [36]. In these countries, vaccine acceptance was positively associated with COVID-19 knowledge and threat regarding COVID-19. However, as information that COVID-19 vaccine-induced death spread within communities [38], a study indicated a low-risk perception of COVID-19 among Sub-Sahara Africans [39]. Therefore, fear of adverse effects from vaccines “makes sense”. Having a perceived low risk of disease can make vaccination less of a requirement. According to the Health Believe Model (HBM), the understanding of diseases enables people to perceive threat and thus, vaccine acceptance increases [28,40,41,42]. With reference to studies carried out in Kenya, Nigeria, Senegal, Turkey, and Thailand [34,43,44,45,46,47], health education was best suited to fill existing knowledge gaps and motivate parents to get their children vaccinated; this was the primary influence on the participants’ decision on vaccination. Thereby, knowledge of VPDs raises parents’ awareness of the healthcare, prevention, and the consequences of diseases for the child if measures are not taken to ensure health. This finding showed that vaccine acceptance increases as parents’ knowledge of VPDs increases. Therefore, the implementation of health education programs may be most beneficial. At the same time, the local system of customs and values (background, tradition, healing concept, etc.) should be considered for people’s understanding and perception of VPDs [Figure 1].

## 5. Limitations of the Study

In this study, some limitations were expected because this is a cross-sectional study, and the sampling method is susceptible to selection bias. Although the sample size was small, it is representative of the population which it resamples as it was calculated using the simple random sampling (SRS) method. Unfortunately, this method lacks the use of available knowledge concerning the population. Only the participants that were present in the district at the time of the interview and that met the survey inclusion criteria were considered in the sample.

## 6. Conclusions

This study identified knowledge of vaccine-preventable diseases as a factor influencing vaccine acceptance among parents in the Foumbot district. The low perceived threat of diseases is a matter of concern as it contributes to vaccine hesitancy. Consequently, vaccine acceptance may or may not only be based directly on a knowledgeable comprehension of the vaccination but also on the understanding of vaccine-preventable disease and the interpretation of the received information in communities [Figure 1]. The findings of this study suggest investing in health education programs to raise parents’ perceived threat towards disease infections and thereby improve preventive health behaviors such as vaccination to avoid diseases. Health education programs should not exclusively be limited to avoiding diseases or how to cope with diseases. It should also consider the local understanding of health issues and diseases. Future research should investigate possibilities in understanding the dynamics of communities regarding health issues while tailoring immunization programs to the local context.

## Figures and Tables

**Figure 1 ijerph-19-02721-f001:**
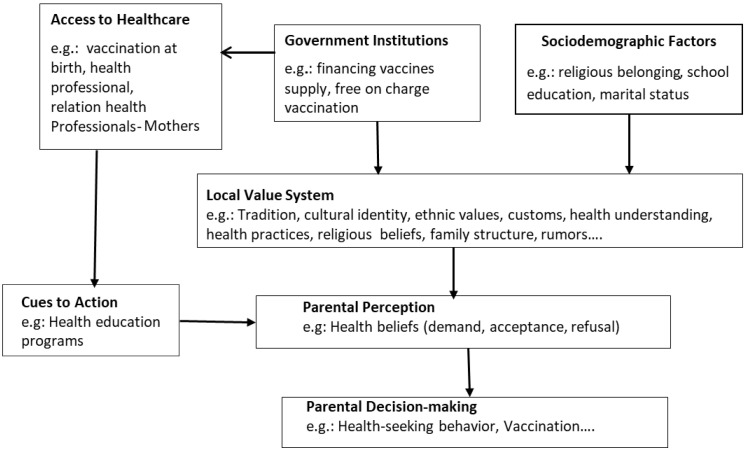
Proposed community-based conceptual framework of parental decision-making.

**Table 1 ijerph-19-02721-t001:** The EPI vaccines coverage of children 12–23 months from this study compared to data for Cameroon from the Sociodemographic Health Survey (DHS) in 2018.

Vaccines	Case	Percentage	Percentage
	Present Study	Present Study	Cameroon
	(*n* = 160)	(*n* = 160)	DHS (2018)
	Case	%	%
BCG	158	98.8	87.1
DPT 1	144	90	83
DPT 2	134	83.8	79
DPT 3	125	78.1	72
OPV 1	143	89.4	86
OPV 2	134	83.8	80
OPV 3	124	77.5	67
VAA (Yellow Fever)	101	63.1	69.3
VAR (Measles)	101	63.1	65
Not vaccinated	2	1.25	10
Completely vaccinated	96	60	52

**Table 2 ijerph-19-02721-t002:** Given reasons by parents for missing the EPI (*n* = 64).

Barriers in Vaccine Acceptance:		Cases	Percentage
Missing Information	Ignores the necessity for vaccination	6	17.2%
	Ignores the necessity of the 2nd and 3rd dose	3	
	Fear of adverse effects	2	
No interest	Discouraged by the entourage	3	17.2%
	Does not trust vaccination	8	
Quality of vaccination services	Absence of the vaccination personnel	1	65.6%
	Vaccines not available	4	
	Long waiting time	3	
	No flexible time schedule	34	

**Table 3 ijerph-19-02721-t003:** Sociodemographic characteristics of participants (*n* = 160) and their associations with complete childhood immunization.

Independent Variables			Complete Vaccination			Bivariate Analysis	
			Yes	No	Total	OR (95% CI)	*p*-Value
			*n* = 96 (%)	*n* = 64 (%)	*n* = 160 (%)		
Maternal Age	<25	Yes	53 (60.9)	34 (39.1)	87 (54.4)	1.09 (0.57–2.05)	0.461
		No	43 (58.9)	30 (41.1)	73 (45.6)		
	25 to 34	Yes	33 (60)	22 (40)	55 (34.4)	1 (0.51–1.94)	0.566
		No	63 (60)	42 (40)	105 (65.6)		
	>34	Yes	10 (55.6)	8 (44.4)	18 (11.3)	0.81 (0.30–2.18)	0.434
		No	86 (60.6)	56 (39.4)	142 (88.8)		
Education	No formal education	Yes	0 (0)	2 (100)	2 (1.3)	0	0.158
		No	96 (60.8)	62 (39.2)	158 (98.8)		
	Primary	Yes	37 (54.4)	31 (45.6)	68 (42.5)	0.67 (0.35–1.26)	0.141
		No	59 (64.1)	33 (35.9)	92 (57.5)		
	Secondary	Yes	57 (64.8)	31 (35.2)	88 (55)	1.55 (0.82–2.94)	0.115
		No	39 (54.2)	33 (45.8)	72 (45)		
	University	Yes	2 (100)	0	2(1.3)		0.36
		No	94 (59.5)	64 (40.5)	15 (98.8)		
Child Birth order	1st	Yes	27 (67.5)	13 (32.5)	40 (25)	1.53 (0.72–3.26)	0.176
		No	69 (57.5)	51 (42.5)	120 (75)		
	2nd	Yes	32 (68.1)	15 (31.9)	47 (29.4)	1.63 (0.79–3.34)	0.121
		No	64 (56.6)	49 (43.4)	113 (70.6)		
	3rd	Yes	14 (50)	14 (50)	28 (17.5)	0.61 (0.26–1.38)	0.164
		No	82 (62.1)	50 (37.9)	132 (82.5)		
	>3	Yes	23 (51.1)	22 (48.9)	45 (28.1)	0.6 (0.29–1.20)	0.105
		No	73 (63.5)	42 (36.5)	115 (71.9)		
Marital Status	Single	Yes	17 (81)	4 (19)	21 (13.1)	3.23 (1.03–10.09)	0.028
		No	79(56.8)	60 (43.2)	139 (86.9)		
	Married	Yes	79 (57.2)	59 (42.8)	138 (86.3)	0.393 (0.13–1.12)	0.06
		No	17 (77.3)	5 (22.7)	22 (13.8)		
	Widow	Yes	0	1 (100)	1 (0.6)	0	0.4
		No	96 (60.4)	63 (39.6)	159 (99.4)		
	Divorced	No	96 (60)	64 (40)	160 (100)		
Religion	Catholic	Yes	12 (75)	4 (25)	16 (10)	2.14 (0.65–6.96)	0.1533
		No	84 (58.3)	60 (41.7)	144 (90)		
	Protestant	Yes	24 (82.8)	5 (17.2)	29 (18.1)	3.93 (1.41–10.94)	0.004
		No	72 (55)	59 (45)	131 (81.9)		
	Muslim	Yes	60 (52.2)	55 (47.8)	115 (71.9)	0.27 (0.12–0.61)	0.001
		No	36 (80)	9 (20)	45 (28.1)		
	Animist	No	96 (60)	64 (40)	160 (100)		
Occupations	Formal sector	Yes	8 (66.7)	4 (33.3)	12 (7.5)	1.36 (0.39–4.73)	0.434
		No	88 (59.5)	60 (40.5)	148 (92.5)		
	Informal Sector	Yes	88 (59.5)	60 (40.5)	148 (92.5)	0.73 (0.21–2.54)	0.434
		No	8 (66.7)	4 (33.3)	12 (7.5)		
Reason for vaccinating	Prevent the occurrence of diseases	Yes	64 (58.7)	45 (41.3)	109 (68.1)	0.84 (0.42–1.67)	0.379
		No	32 (62.7)	19 (37.3)	51 (31.9)		
	Decrease the severity of diseases	Yes	1 (50)	1 (50)	2 (1.2)	0.66 (0.04–10.79)	0.641
		No	95 (60.1)	63 (39.9)	158 (98.8)		
	No idea	Yes	31 (63.3)	18 (36.7)	49 (30.6)	1.2 (0.61–2.4)	0.35
		No	65 (58.6)	46 (41.4)	111 (69.4)		
Knowledge of VPD and communicable diseases	Good knowledge	Yes	88 (63.7)	50 (36.2)	138 (86.2)	3.08 (1.21–7.85)	0.01
		No	8 (36.3)	14 (63.6)	22 (13.8)		
	Poor knowledge	Yes	8 (36.4)	14 (63.6)	22 (13.8)	0.32 (0.12–0.82)	0.014
		No	88 (63.8)	50 (36.2)	138 (86.3)		
Age-appropriate vaccination	At Birth	Yes	83 (64.3)	46 (35.7)	129 (80.6)	2.5 (1.12–5.55)	0.02
		No	13 (41.9)	18 (58.1)	31 (19.4)		
	No idea	Yes	11 (44)	14 (56)	25 (15.6)	0.4 (0.19 + 1.09)	0.06
		No	85 (63)	50 (37)	135 (84.3)		
	Other date	Yes	2 (33.3)	4 (66.6)	6 (3.7)	0.32 (0.06–1.79)	0.17
		No	94 (60)	60 (40)	154 (96.9)		
Be aware of the vaccination schedule	Be aware	Yes	29 (80.6)	7 (19.4)	36 (22.5)	3.52 (1.43–8.65)	0.002
		No	67 (54)	57 (46)	124 (77.5)		
Retention of the vaccination booklet	Keep	Yes	88 (62)	54 (38)	142 (88.8)	2.04 (0.76–5.48)	0.121
		No	8 (44.4)	10 (55.6)	18 (11.3)		

**Table 4 ijerph-19-02721-t004:** Information criteria of multivariate logistic regression analysis of vaccination decision-making in parents with children between 12 to 23 months (*n* = 160).

Model	Action	Effect(s)	Model Fitting Criteria			Effect Selection Tests		
			AIC	BIC	−2 Log-Likelihood	Chi-square ^a^	df.	Sig.
Step 0	Entered	Constant	85.70	88.78	83.70	.		
Step 1	Entered	Religion	76.58	82.73	72.58	11.12	1	0.001
Step 2	Entered	Age-appropriate vaccination	67.74	76.97	61.74	10.84	1	0.001
Step 3	Entered	Knowledge of VPDs	64.16	76.46	56.14	5.58	1	0.018

Stepwise Method: Forward Selection. The Chi-Square for entry is based on the likelihood ratio test.

**Table 5 ijerph-19-02721-t005:** Model fitting information of multivariate logistic regression analysis of vaccination decision-making in parents with children between 12 to 23 months (*n* = 160).

Model	Model Fitting Criteria			Likelihood-Ratio-Tests			AUC
	AIC	BIC	−2 Log-Likelihood	Chi-Square	df.	Sig.	
Only Intercept	85.70	88.78	83.70				
Final	64.16	76.46	56.16	27.54	3	0.0001	0.73

## Data Availability

The datasets used and/or analyzed during the current study are available from the corresponding author on reasonable request.

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
