# Peer review of "Insights into Predictors of Vaccine Hesitancy and Promoting Factors in Childhood Immunization Programs—A Cross-Sectional Survey in Cameroon"

_ijerph, 2022, doi:10.3390/ijerph19052721_

Round 1
Reviewer 1 Report
This is a well-presented study of general interest in terms of public health and vaccination policy. I have a few comments which may improve the MS.
Line 28: define EPI on first use
The referencing needs to be sorted out e.g. line 42 :162729
All references need to be in [ ] e.g. line 44
Line 47; is 221021 a date or a reference?
Table 3: vaccination personnel (not personal)
Line 230: I did not understand this quotation? specifically 'bait out?'
Author Response
Response to Reviewer 1 Comments
Point 1: Line 28: define EPI on first use.
Response 1: This has been added
Point 2: The referencing needs to be sorted out e.g. line 42 :162729
Response 2: This was due to formating. It has been corrected
Point 3: All references need to be in [ ] e.g. line 44
Response 3: This was due to formating. It has been corrected
Point 4: Line 47; is 221021 a date or a reference?
Response 4: There are references but the error was due to formating. It has been corrected
Point 5: Table 3: vaccination personnel (not personal)
Response 5: This has been corrected
Point 6: Line 230: I did not understand this quotation? specifically 'bait out?'
Response 6: The quotation has been deleted.
Reviewer 2 Report
In this study, children aged 12-23 months were surveyed to investigate the current situation of vaccination hesitancy, identify the factors influencing vaccination hesitancy and propose corresponding countermeasures. Please correct the style and format of this articles.
Background
The background of the study is incomplete, please add what is vaccine hesitancy? What are the possible effects of vaccine hesitancy? Please describe the study's vaccination status, willingness to vaccinate, and the current characteristics of vaccination? Why is this vaccine the subject of the study? What is the significance of the study?
Methods
How is vaccine hesitancy determined? Please describe in detail the indicators used in the questionnaire.
Results
Please add the results of the logistic regression analysis including the factors influencing vaccine hesitancy.
Author Response
Point 1: Background
Response 1: The background has been completed according to the comment.
Point 2: Method
Response 2: The missing details has been added
Point 3: Results
Response 3: The results of the logistic regression analysis have been added.and also the factors influencing vaccine hesitancy/willingness.
Reviewer 3 Report
This study aims to identify the predictor factors of vaccination failure and the factors that promote vaccination among children between the ages of 12 to 23 months in the Foumbot district, West Cameroon. To achieve these goals, a total of 160 mothers of children, aged between12 to 23 months, were selected by simple random sampling technique, using a pre-tested structured questionnaire. The outcome of this survey shows that 60% of the children studied were completely vaccinated, 37.75% were partially vaccinated, and 1.25% had not received any vaccine.
The short form should be preceded by the extension form. Please, check all text (i.e. EPI in the abstract, or WHO,...).
Please, check all references: they are inserted as a number without brackets.
Line 42 :" vaccine hesitancy among parents was the main reason for outbreak 162729", please clarify which means 162729.
The aims of the study should be written clearly. The authors stated, "This study identifies and examines the predictors of vaccine hesitancy and the factors promoting vaccine acceptance among children between the ages 12 to 23 months in the Foumbot district, West Cameroon." This sentence seems a result. Please, re-write it.
Please check lines 63, 106, 125, 145 (Error! Reference source not found).
In the socio-economic factors, the authors missed inserting the economic factors (i.e. income range).
The results section is very confused: several tables are inserted without a description of the results.
The limitations of the study should be extended: more other limitations are present (the number of interviewed subjects, the absence of income information, etc..).
The conclusion should be re-written. The authors stated: "The rate of vaccine hesitancy was higher among Muslims", nevertheless, it is influenced by the enrolled subjects (115/160 were Muslims). Please, check the data and re-write the conclusion section.
Author Response
Response to Reviewer 3 Comments
Point 1: The short form should be preceded by the extension form. Please, check all text (i.e. EPI in the abstract, or WHO,...).
Response 1: This has been completed according to the comment.
Point 2: Please, check all references: they are inserted as a number without brackets
Response 2: This has been corrected. This was due to the formating
Point 3: Line 42 :" vaccine hesitancy among parents was the main reason for outbreak 162729", please clarify which means 162729
Response 3: They are references. The error happened due to the formating. We corrected it.
Point 4: The aims of the study should be written clearly. The authors stated, "This study identifies and examines the predictors of vaccine hesitancy and the factors promoting vaccine acceptance among children between the ages 12 to 23 months in the Foumbot district, West Cameroon." This sentence seems a result. Please, re-write it.
Response 4: The background has been modified accordingly
Point 5: Please check lines 63, 106, 125, 145 (Error! Reference source not found).
Response 5: Error due to formating. This has been corrected.
Point 6: In the socio-economic factors, the authors missed inserting the economic factors (i.e. income range).
Response 6: Since we focus on sociodemogaphic factors, income range was not considered. Occupation was considered instead of income range.This is one limitation of this kind of study.
Point 7: The results section is very confused: several tables are inserted without a description of the results.
Response 7: This has been corrected. The results are now detailed in the results section
Point 8: The limitations of the study should be extended: more other limitations are present (the number of interviewed subjects, the absence of income information, etc..).
Response 8: The limitation of the study has been extended to the number of participant and also the lack in the sampling method.
Point 9: The conclusion should be re-written. The authors stated: "The rate of vaccine hesitancy was higher among Muslims", nevertheless, it is influenced by the enrolled subjects (115/160 were Muslims). Please, check the data and re-write the conclusion section.
Response 9: The conclusion has been re-written while considering the comment.
Round 2
Reviewer 3 Report
Based on the reviewers’ comments, the authors have improved their manuscript. Nevertheless, I suggest several modifications before endorsing the publication.
The title should be modified: it is very important that the authors clarify that the study refers to the African situation.
The Abstract section should be revised. Based on the authors’ guidelines (https://www.mdpi.com/journal/ijerph/instructions), “The abstract should be a total of about 200 words maximum”, while it is more than 350 words. Please, reduce it.
The keywords should be reviewed focusing on the thematic of the manuscript.
The introduction section should be improved. I suggest focusing on the thematic presenting, for example, the state of the art of vaccination in the same geographical area where the study was performed.
Section 3 should be re-named: it is the “results” section. Moreover, the table should be inserted in the specific point of the text (authors’ guidelines “All Figures, Schemes, and Tables should be inserted into the main text close to their first citation and must be numbered following their number of appearance”). Moreover, “All Figures, Schemes, and Tables should have a short explanatory title and caption”: please, insert a short title. Furthermore, the description of each subsection should be clear and concise presenting the results clearly.
Both in the introduction and in the discussion, in order to improve the attractiveness of the paper, several considerations about the anti-COVID-19 vaccination should be inserted with the relative efforts to reduce the vaccine hesitancy. In this regard, you can read and cite the following references: DOI: 10.3390/vaccines9111291; DOI: 10.3390/vaccines9121415; DOI: 10.3390/diagnostics11060955; DOI: 10.3390/jcm10245876.
Author Response
Response to Reviewer 3 Comments
Point 1: The title should be modified: it is very important that the authors clarify that the study refers to the African situation.
Response 1: The title has been modified accordingly. It refers now to Cameroon.
Point 2: The Abstract section should be revised. Based on the authors’ guidelines (https://www.mdpi.com/journal/ijerph/instructions), “The abstract should be a total of about 200 words maximum”, while it is more than 350 words. Please, reduce it.
Response 2: The Abstract has been reduced accordingly (Now 198 words).
Point 3: The introduction section should be improved. I suggest focusing on the thematic presenting, for example, the state of the art of vaccination in the same geographical area where the study was performed.
Response 3: The introduction has been re-written accordingly.
Point 4: Section 3 should be re-named: it is the “results” section. Moreover, the table should be inserted in the specific point of the text (authors’ guidelines “All Figures, Schemes, and Tables should be inserted into the main text close to their first citation and must be numbered following their number of appearance”). Moreover, “All Figures, Schemes, and Tables should have a short explanatory title and caption”: please, insert a short title. Furthermore, the description of each subsection should be clear and concise presenting the results clearly.
Response 4: Section 3 has been renamed as “Results”. The tables and figures have been inserted according to the “authors’ guideline”. The results are also presented.
Point 5: Both in the introduction and in the discussion, in order to improve the attractiveness of the paper, several considerations about the anti-COVID-19 vaccination should be inserted with the relative efforts to reduce the vaccine hesitancy. In this regard, you can read and cite the following references: DOI: 10.3390/vaccines9111291; DOI: 10.3390/vaccines9121415; DOI: 10.3390/diagnostics11060955; DOI: 10.3390/jcm10245876.
Response 5: Elements and considerations about anti-COVID 19 vaccination have been inserted in both introduction and discussion.